# Circulating Lymphocytes Reflect the Local Immune Response in Patients with Colorectal Carcinoma

**DOI:** 10.3390/diagnostics12061408

**Published:** 2022-06-07

**Authors:** Johanna Waidhauser, Pia Nerlinger, Florian Sommer, Sebastian Wolf, Stefan Eser, Phillip Löhr, Andreas Rank, Bruno Märkl

**Affiliations:** 1Department of Hematology and Oncology, Medical Faculty, University of Augsburg, 86156 Augsburg, Germany; pia.nerlinger@uk-augsburg.de (P.N.); phillip-loehr@uk-augsburg.de (P.L.); andreas.rank@uk-augsburg.de (A.R.); 2Department of General, Visceral and Transplant Surgery, Medical Faculty, University of Augsburg, 86156 Augsburg, Germany; florian.sommer@uk-augsburg.de (F.S.); sebastian.wolf@uk-augsburg.de (S.W.); 3Department of Gastroenterology and Infectious Diseases, Medical Faculty, University of Augsburg, 86156 Augsburg, Germany; stefan.eser@uk-augsburg.de; 4General Pathology and Molecular Diagnostics, Medical Faculty, University of Augsburg, 86156 Augsburg, Germany; bruno.maerkl@uka-sience.de

**Keywords:** circulating lymphocytes, colorectal carcinoma, tumor immune response, flow cytometry, tumor-infiltrating lymphocytes, lymph node size

## Abstract

Tumor-infiltrating lymphocytes (TILs) correlate with the number and size of the surrounding lymph nodes in patients with colorectal carcinoma (CRC) and reflect the quality of the antitumor immune response. In this prospective study, we analyzed whether this response correlated with the circulating lymphocytes in peripheral blood (PB). In 47 patients with newly diagnosed CRC, flow cytometry was performed to analyze the B cells, T cells, NK cells, and a variety of their subsets in PB. The results were correlated with TILs in the resected tumor and with the number and size of the surrounding lymph nodes in nodal negative (N- patients (LN5: number of lymph nodes measuring ≥5 mm) and the metastasis-to-lymph node size ratio (MSR) in nodal positive patients (N+). Differences between the number of TILs could be seen between N+ and N- patients, dependent on the LN5 and MSR categories, with higher values in N- cases and in patients with a higher LN5 category or a lower MSR. Additionally, higher values of various circulating lymphocyte subgroups were observed in these patients. For the total PB lymphocytes, CD8 cells, and some of their subgroups, a positive correlation with the TILs was found. This study shows that circulating lymphocytes—in particular, cytotoxic T cells—correlate with the local antitumor immune response displayed by TILs and lymph node activation. Our findings indicate that local and generalized antitumor immune responses are concordant with their different components.

## 1. Introduction

Colorectal carcinoma (CRC) still comprised the third-most frequent type of cancer and the second-leading cause of death worldwide in 2020 [1]. Early tumor detection is crucial for long-time prognosis, and different types of screening exist, from which colonoscopy and stool-based examinations of occult fecal blood are the best evaluated options [2]. Besides detection in an early stage, the local immune response to tumor tissue in the form of tumor-infiltrating lymphocytes (TILs) plays an important role in controlling tumor growth, the formation of metastasis, and the recurrence in patients with colorectal carcinoma (CRC). A high intra-tumoral density of infiltrating CD3+ and CD8+ lymphocytes was associated with a smaller local tumor extension, reduced likelihood of metastases, and a better prognosis [3,4,5]. In addition to tumor-infiltrating lymphocytes, the local lymph node size and number seem to be associated with endogenous tumor control. A histopathological examination of at least 12 lymph nodes is recommended in several national guidelines [6,7,8]. A higher lymph node yield was associated with a better survival rate of patients with stage II or III colon cancer [9]. The underlying mechanism, however, remains controversial [10]. Stage migration theory (Will–Rogers phenomenon) with a higher likelihood of missed lymph node metastases in patients with a lower lymph node harvest, resulting in inadequate cancer therapy, is a longstanding possible explanation for this phenomenon [11]. However, in recent years, another thesis has gained prominence: a stronger immune reaction to the tumor results in a higher amount and greater size of the surrounding lymph nodes, associated with a higher lymph node yield and better prognosis (Figure 1) [12,13,14,15,16]. A retrospective analysis of our research group showed that the number and size of the surrounding lymph nodes in nodal negative (N-) patients correlates with the number of TILs [17]. Additionally, a similar association was found for nodal-positive (N+) patients—a lower metastasis-to-lymph node size ratio (MSR), which correlates with a higher number of immune cells in the lymph node, was also associated with a better outcome [18]. Whether these expressions of the antitumor immune response are also displayed by alterations of the peripheral blood immune status has not yet been sufficiently evaluated. Preexisting studies concentrated on the impact of circulating lymphocytes on the survival of CRC patients and their differences from healthy controls [19,20,21,22] or on select subsets, such as NK cells [23] or regulatory CD4+ cells [24].

The aim of this study is to prospectively confirm the association between TILs and lymph node size and number and to investigate a possible correlation with circulating lymphocyte subsets to show that both parameters (local and generalized immune response) are part of a strong antitumor immune response, a further argument against the Will–Rogers thesis in patients with CRC.

## 2. Materials and Methods

Study population and trial design. Patients with newly diagnosed CRC who were scheduled for surgery at the University Medical Center Augsburg between December 2018 and November 2020 with no history of chronic infectious disease or inherent or acquired immunodeficiency or autoimmune disorder were included. Surgery was planned for tumor stages UICC I–III. Additionally, a small number of patients with unknown UICC stage IV prior to surgery were included. The study was approved by the medical ethical committee of Ludwigs Maximilians University Munich (reference number 18-726), and written informed consent was obtained from all patients.

Histological assessment of tumor-infiltrating lymphocytes and surrounding lymph nodes. The resected colon was immediately delivered to the Institute of Pathology and Molecular Diagnostics of the University Medical Center Augsburg. Standard histological characteristics, such as tumor size, histological classification, tumor budding, mismatch repair (MMR) status (using immunohistochemistry with an expression analysis of PMS2 and MSH6), and, depending on the UICC stage, the BRAF and panRAS mutational status, were assessed. To optimize the lymph node yield, methylene blue was injected into an artery as described in previous studies [25,26,27]. The goal was to obtain a minimum of 25 lymph nodes per case. All lymph nodes were carefully screened for tumor metastases based on at least two HE step sections. The longitudinal extent was measured, and N- samples were categorized as ≥5 mm or <5 mm (LN5 category). Cases with 0 to 1 lymph nodes ≥5 mm were designated as LN5 very low, cases with 2–5 lymph nodes ≥5 mm as LN5 low, and cases with more than 5 lymph nodes ≥5 mm as LN5 high [15,17]. For tumor-affected lymph nodes, the MSR was calculated according to the following formula, as previously published [18]:(1)$$\sum_{i=1}^{m}xi/\sum_{k=1}^{n}yk,$$
where *xi* = maximum diameter of the tumor infiltrate in mm, *m* = total number of metastasized nodes per case, *yk* = maximum diameter of lymph node in mm, and *n* = total number of lymph nodes per case. Cases with a MSR ≥ 0.1070 were classified as group 0, and cases with <0.1070 were classified as group 1.

For the detection of tumor-infiltrating lymphocytes, immunohistochemical staining with antibodies against CD3 (2GV6 rabbit monoclonal antibody; 1:100; Roche Ventana Medical Systems, Tucson, AZ, USA) and CD8 (144B mouse monoclonal antibody; ready to use; Cell Marque, Rocklin, CA, USA) was used. All immunohistochemical reactions were performed on an automatic platform using the Ultraview DAB detection kit (Benchmark Ultra, Roche Diagnostics, Mannheim, Germany). All slides were then digitalized with a Philips Ultrafast Scanner with a 20× lens (Philips Health Systems, Hamburg, Germany). Using the implemented Image Management System, the two corresponding slides of each case were synchronized, and representative areas of two different localizations at the tumor center (TC) and invasive front (IF) with a high and intermediate infiltration of CD3+ or CD8+ cells were selected by an experienced pathologist (BM) and exported as image files that served as the basis for digital counting.

Cell counting was performed using digital quantifier software (VMscope, Berlin, Germany). For all cases, two different localizations at the TC and invasive front IF with a high and intermediate infiltration of CD3+ or CD8+ cells were analyzed, resulting in eight different values for each patient. The high and intermediate infiltration rates for each cell type and localization were then summarized as one mean value, resulting in four categories of TILs for each patient (Figure 2).

Analysis of lymphocytes and subsets. Peripheral blood lymphocytes were measured as previously published (i.e., by our study group) [28,29,30]. Blood samples (EDTA blood) were taken prior to surgery to perform flow cytometry (FC500 from Beckman Coulter, Brea, CA, USA) in our local laboratory within 24 h. The flow cytometric panel was established using a control group of 50 healthy volunteers (median age 43 years, 34% women), mostly blood donors from the blood bank of the University Medical Center Augsburg [29,30]. Cell staining was done using commercial fluoreszeinisothiocyanat (FITC-), phycoerythrin (PE-), phycoerythrin Texas red-X (ECD-), and phycoerythrin-cyanin (PC5 and PC7)-labeled antibodies purchased from Beckman Coulter (Brea, CA, USA) and Biolegend (San Diego, CA, USA). Initial results for the lymphocyte subsets were given as percentages. Absolute values were calculated using the absolute leukocyte counts measured with Stem-Count (Stem-Kit, Beckman Coulter). Analysis of the different lymphocyte subsets was done as previously reported by our research group [28,29,30].

B lymphocytes were identified by the presence of CD19 (CD19-PC7 IM3628) and were further divided into naïve (IgD+ CD27-; IgD-FITC B30652, CD27-ECD B26603); memory (IgD+ CD27+); class-switched memory (IgD- CD27+); and transitional (CD24hi CD38hi; CD24-PE IM1428U, CD38-PC5 A07780) subsets. 

The total T-lymphocyte counts were obtained using CD3 and Stem-Count, as described above. For further T-cell analyses, lymphocytes were identified using forward and side scatter. T-lymphocytes were defined by positivity for CD8 or CD4 and were subdivided into naïve (CD62L+ CD45RA+) and memory T cells (CD4+ CD45RA- CD45RO+/CD8+ CD45RA- CD45RO+), which were further divided into central memory (CD62L+ CD45RA-), effector memory (CD62L- CD45RA), effector memory RA+ (EMRA; CD62L- CD45RA+), and activated CD4+ or CD8+ cells (HLA-DR+ or CD69+) and regulatory cells (CD4+ CD25hi, displaying IL2R+ CD4+ cells). 

Furthermore, types 1, 2, and 17 CD4+ T-helper (Th1, Th2, and Th17) cells were identified using antibodies against CXCR3, CCR4, CCR5, and CCR6. Th1 cells were defined as CD4+ CXCR3+ CCR4- CCR5+ CCR6-, Th2 cells as CD4+ CXCR3- CCR4+ CCR5- CCR6-, and Th17 cells as CD4+ CXCR3- CCR4+ CCR5- CCR6+.

Within cytotoxic CD8+ T lymphocytes, activated subsets of the early (CD28+ CD27+), intermediate (CD28- CD27+), and late (CD28- CD27-) statuses were identified, along with exhausted (CD279+) and terminal effector (CD279- CD57+) cells. NK-like T cells were defined as CD56+ CD3+.

NK lymphocytes were detected as CD56+ cells and subdivided into 3 NK subsets (CD56+ CD16+, CD56dim CD16bright, and CD56bright CD16dim).

Detailed information regarding the antibodies and gating strategy are displayed in Appendix A.

Statistical analysis. The results of the descriptive analysis were given as median values and interquartile ranges. A Mann–Whitney *U* test was performed to compare independent samples. Spearman’s rank correlation coefficient was calculated to analyze potential correlations between circulating and tumor-infiltrating lymphocytes. *p*-values < 0.05 were considered statistically significant. Data were analyzed with SPSS for Windows (IBM SPSS Statistics 24, Armonk, NY, USA). Figures were designed using SPSS or R 4.0.2.

## 3. Results

Population characteristics. A total of 47 patients with newly diagnosed CRC were included. The median age was 66 years (range: 42–84 years), and 29 patients were male. Patients at all UICC stages were included. UICC IV patients were scheduled for surgery unaware of the metastatic situation. The median lymph node harvest was 35, and 16 patients were positive for metastases. Thirty-two patients had a right-sided tumor (coecum, ascending colon, and transverse colon up to splenic flexure), and microsatellite instability was found in 11 patients (Table 1). The median values and interquartile ranges for the peripheral blood and tumor-infiltrating lymphocytes of the entire cohort are displayed in Table 2.

Lymph node-negative versus -positive patients. Regarding tumor-infiltrating lymphocytes in patients with or without lymph node affection, higher values of CD3 and CD8 cells in the TC and in the IF could be observed for N- patients. The differences were significant for CD8 lymphocytes in the IF (621/mm^2^ vs. 351/mm^2^; *p* = 0.041; Figure 3A). An analysis of the circulating lymphocyte subsets revealed no differences for B cells, T cells, NKs cells, or any of their analyzed subsets.

Lymph node-negative patients. Of the 31 lymph node-negative patients, seven were categorized in the LN5 very low category (0 to 1 lymph nodes with a diameter ≥ 5 mm), 11 were in the LN5 low category (2–5 lymph nodes ≥ 5mm), and 13 were in the LN5 high group (>5 lymph nodes ≥ 5 mm). Comparing the tumor-infiltrating lymphocytes between the three LN5 groups showed higher TILs values for all four categories for LN5 low patients compared to LN5 very low patients, as well as for CD8 cells in the TC and CD3 and CD8 cells in the IF between LN5 very low and LN5 high patients. A statistically significant difference, however, was only seen for CD8 cells in the TC between LN5 very low and LN5 low patients (Figure 3B). Regarding circulating lymphocytes, significantly lower values of the total B cells (82/µL vs. 127/µL; *p* = 0.037) and their subgroups of naïve (51/µL vs. 92/µL; *p* = 0.036), memory (2/µL vs. 8/µL; *p* = 0.019), and class-switched B cells (7/µL vs. 17/µL; *p* = 0.024) were seen in the LN5 very low group compared to the LN5 high group. Between the LN5 very low and LN5 low groups, significantly higher values of regulatory CD4 cells represented by CD4+CD25 high cells (18/µL vs. 15/µL; *p* = 0.027) were seen for the LN5 very low category. Effector memory CD4 cells (145/µL vs. 95/µL; *p* = 0.044) and CD69+ CD4 cells (15/µL vs. 11/µL; *p* = 0.008) showed higher values in the LN5 low group compared to the LN5 very low group.

Lymph node-positive patients. Patients with lymph node affection (*n =* 16) were divided into two groups according to their MSR. Patients with a high proportion of metastasis compared to the total lymph node size were classified as MSR0 (*n =* 10), and patients with a smaller relative metastasis size were classified as MSR1 (*n =* 6). TILs in the MSR1 group were higher for CD8 TC (249/mm^2^ vs. 168/mm^2^; *p* = 0.147), CD3 IF (1259/mm^2^ vs. 867/mm^2^; *p* = 0.220), and CD8 IF (538/mm^2^ vs. 302/mm^2^; *p* = 0.428), although the differences did not reach statistically significant *p*-values (Figure 3C). Comparing circulating lymphocytes between the two groups revealed significantly higher values in the MSR1 group for the total lymphocytes (1268/µL vs. 1711/µL; *p* = 0.31), CD3 cells (768/µL vs. 1183/µL; *p* = 0.22), HLADR+ CD4 cells (35/µL vs. 67/µL; *p* = 0.005), CD8 cells (195/µL vs. 402/µL; *p* = 0.031), and their subsets of memory CD8 cells (63/µL vs. 158/µL; *p* = 0.003), central memory CD8 cells (18/µL vs. 45/µL, *p* = 0.016), effector memory CD8 cells (69/µL vs. 182/µL; *p* = 0.031), EMRA CD8 cells (31/µL vs. 164/µL; *p* = 0.031), intermediate CD8 cells (5/µL vs. 38/µL; *p* = 0.007), late CD8 cells (34/µL vs. 220/µL; *p* = 0.031), exhausted CD8 cells (47/µL vs. 126/µL; *p* = 0.022), and HLADR+ CD8 cells (38/µL vs. 154/µL; *p* = 0.042).

Combined LN5 and MSR category. Combining the LN5 very low and MSR0 groups into a LN5/MSR0 group, representing patients with a lower nodal immune response, and combining the LN5 low, LN5 high, and MSR1 groups into a LN5/MSR1 group revealed significantly higher values of TILs for CD8 TC (167/mm^2^ vs. 328/mm^2^; *p* = 0.002) and CD3 IF (895/mm^2^ vs. 1340/mm^2^; *p* = 0.009) in the LN5/MSR1 group (Figure 3D). Additionally, significantly higher values of the circulating total lymphocytes (1386/µL vs. 1070/µL; *p* = 0.046), class-switched B cells (19/µL vs. 8/µL; *p* = 0.011), total T lymphocytes (955/µL vs. 718/µL; *p* = 0.015), CD69+CD4 cells (16/µL vs. 11/µL; *p* = 0.010), total CD8 cells (260/µL vs. 184/µL; *p* = 0.027), EMRA CD8 cells (76/µL vs. 25/µL; *p* = 0.030), intermediate CD8 cells (17/µL vs. 7/µL; *p* = 0.002), exhausted CD8 cells (71/µL vs. 46/µL; *p* = 0.019), and HLADR+CD8 cells (85/µL vs. 39/µL; *p* = 0.027), as well as NK-like T cells (50/µL vs. 21/µL; *p* = 0.049) and CD56brightCD16dim NK cells (12/µL vs. 9/µL; *p* = 0.030), were seen in the LN5/MSR1 group (Figure 4).

Lymph node harvest. There were no significant differences in the number of lymph nodes obtained between N+ and N- patients, between the MSR0 and MSR1 groups, or between the different LN5 groups (except for the difference between the LN5 low and LN5 high groups, with 31 vs. 41 lymph nodes, respectively; *p* = 0.018).

Microsatellite instability status. MSI tumors were associated with higher values of TILs compared to MSS tumors (significant differences for CD3 TC with 748/mm^2^ vs. 1181/mm^2^; *p* = 0.031), whereas circulating lymphocytes showed no differences according to the MSI status (Figure 3E).

Correlation between tumor-infiltrating and circulating lymphocytes. Correlating the different TILs subgroups with peripheral blood lymphocytes showed a positive correlation between CD8 cells in the TC and total circulating lymphocytes (*p* = 0.019), circulating CD8 cells (*p* = 0.032), and EMRA CD8 cells (*p* = 0.026). Additionally, a positive correlation was seen between CD8 cells in the IF and peripheral blood intermediate CD8 cells (*p* = 0.029). A negative correlation was found between CD8 cells in the IF and naïve B cells (*p* = 0.043; Figure 5).

## 4. Discussion

In this prospective study, we confirmed prior findings from retrospective analyses revealing an association between the number of TILs and lymph node size in N- CRC patients [17]. Additionally, we observed a similar effect for N+ patients, for whom a lower MSR, which displays a stronger local immune response to the tumor, was linked to higher numbers of TILs. However, the differences were not statistically significant, most likely due to the relatively low number of patients in this group. Combining the two subgroups of N+ and N- patients and splitting them into higher and lower nodal immune response groups (LN5 very low or a high MSR vs. LN5 low/high and a low MSR) revealed significant differences in TILs (Figure 3D), benefitting the higher nodal immune response group. Consistent with prior findings, we also observed that patients with affected lymph nodes had lower values of TILs than those without lymph node affection [31].

Few analyses have yet been published regarding correlations between the immunological tumor microenvironment and peripheral blood lymphocytes. In a small study on 18 CRC patients, Hagland et al. found a positive correlation between tumor-infiltrating CD3+ and CD8+ cells in the TC and IF and circulating CD3+ cells, especially CD4+ cells [30]. For CD8+ cells in the peripheral blood, no correlation with TILs was found [32]. Our data confirmed a positive correlation between circulating and tumor-infiltrating lymphocytes. In contrast to the results published by Hagland et al., peripheral blood CD8+ cells were the subgroup with the strongest correlation. The fact that higher values of circulating lymphocytes are associated with higher values of TILs and that, additionally, patients with a stronger immunological lymph node activation show higher values of total peripheral blood T lymphocytes, CD8+ cells, and a variety of their subsets underlines the thesis of a synergistic reaction of the different immunological divisions. The fact that the circulating lymphocyte subsets did not differ between LN- negative and LN- positive patients could be seen as contradictory to the above-described correlations between the local and systemic immune responses. However, in our hypotheses, the immune response plays an essential role in the colon cancer extent, regardless of the stage, and also, a node-positive case might induce a strong immune response and is then associated with a better prognosis compared to low immune response cases.

That the local immune response to tumor cells is crucial for the prognosis of CRC patients has been demonstrated by various study groups. Galon et al. identified tumor-infiltrating immune cells as a better predictor for the survival of CRC patients than the classical histopathological characteristics [3,33], and these findings have been confirmed in a large number of patients [34,35]. Recently, mesenteric lymphocytes, especially higher numbers of CD8+ cells and some of their subsets, were also associated with favorable CRC risk factors, such as a lower depth of invasion [36]. The results of our study also revealed the lymph node size in N- patients [15] and MSR in N+ patients [18] as prognostic markers.

The fact that patients with a higher nodal immune response and higher values of tumor-infiltrating lymphocytes have a better prognosis is another important indication that stage migration theory is not the underlying explanation for a better outcome in patients with a higher lymph node harvest. In contrast, our results suggest, once again, that the local tumor immune response is concordant in its different components, resulting in either a high tumor immune response with larger lymph nodes and higher numbers of TILs and peripheral blood lymphocytes or, in smaller lymph nodes, a higher proportion of tumor cells in metastatic lymph nodes and lower numbers of TILs and circulating lymphocytes in one patient. This supports the theory of “high immune response” or “low immune response” patients.

The lymph node harvest, however, did not show differences between the MRS0 and MSR1 groups or between N+ and N- patients in our study. An explanation for this could be the application of the methylene blue-assisted lymph node dissection technique in resected colorectal specimens, which leads to a higher lymph node yield independent of the lymph node size [13,27]. The lymph node harvest in our study population was considerably higher, with a median of 35 obtained lymph nodes—greater than the required 12 lymph nodes. Between the LN5 low and LN5 high subgroups, however, the lymph node harvest did show significant differences, with 31 and 41 lymph nodes harvested, respectively, indicating that, despite the application of the methylene blue technique, a certain impact of lymph node size on the lymph node harvest remains.

One characteristic of CRC that is at least partly responsible for a higher local tumor immune response is microsatellite instability, which results in a higher mutational burden [37]. Dividing our study population into two groups according to their microsatellite instability status revealed higher numbers of TILs, especially CD3 cells in the TC for MSI-high patients, which is consistent with prior findings [38,39,40] and displays the higher immunogenicity of this subgroup of CRCs. This better immune response was associated with a higher rate of N- cases and a lower rate of distant metastases and resulted in a favorable outcome in previous studies [41,42].

Our study had some limitations. First, the number of patients was relatively small, leading to even smaller subgroups in the different lymph node categories. However, we could detect significant differences in the TILs according to lymph node size and MSR. Another limitation regarding the correlation between tumor-infiltrating and circulating lymphocytes was that a tumor-specific orientation of circulating lymphocytes could only be assumed and not proven in our study. Depth paired analyses of the TCR repertoire of circulating and tumor infiltrating T lymphocytes can be performed in further studies to identify tumor-relevant cells for the characterization and development of markers for tracking in the peripheral blood [43,44].

In summary, we were able to prospectively confirm previous findings regarding higher values of TILs in relation to local lymph node activation in N+ and N- CRC patients. Additionally, we were able to show that this relationship can also be seen in peripheral blood lymphocytes. Altogether, our findings constitute another blow against the stage migration theory. Taking together the prognostic benefit of patients with higher rates of TILs [3,4,5,45] and larger tumor-surrounding lymph nodes [17,18] with our findings, a prognostic role of circulating lymphocyte subsets can be postulated. To investigate this hypothesis, however, further studies with regards to patient outcomes are needed.

## Figures and Tables

**Figure 1 diagnostics-12-01408-f001:**
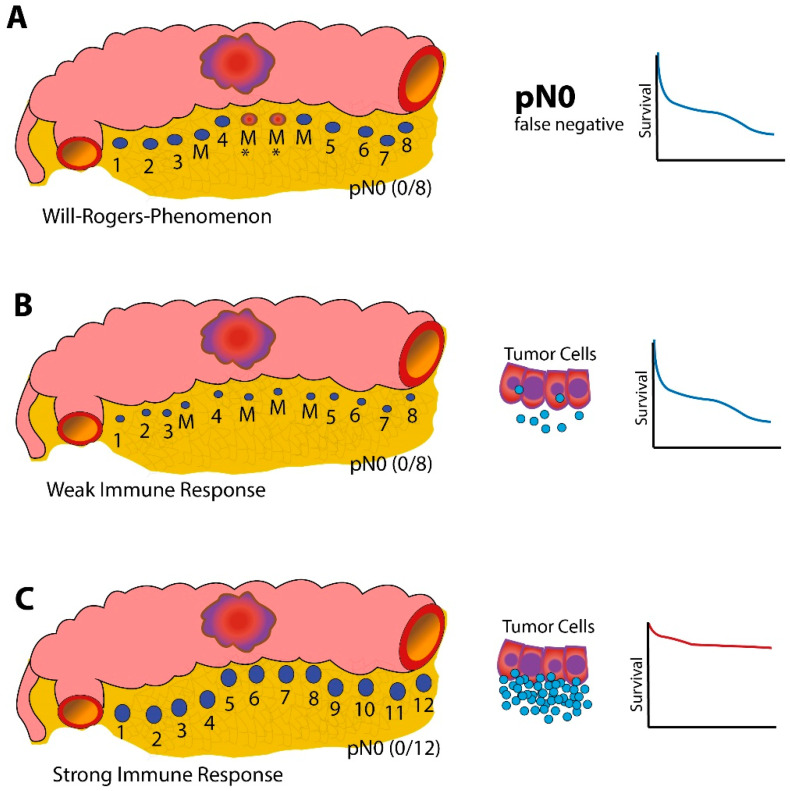
Hypotheses for the prognostic effect of the lymph node yield in colorectal cancer. (**A**) Stage migration effect (Will–Rogers Phenomenon). According to this thesis, the adverse prognostic effect of a low LN yield results from missing lymph node metastases due to insufficient dissection. (**B**) Alternative Thesis. Small hypoplastic lymph nodes are difficult to identify and lead to a low lymph node yield. A small lymph node size correlates with a low density of tumor-infiltrating lymphocytes (TILS) and indicates a weak immune response that enables the immune escape of tumors. (**C**) Large lymph nodes are easy to identify in high numbers and indicate a strong immune response. They are associated with a high density of TILs and a favorable prognosis. Numbers indicate detected local lymph nodes. M = missed lymph nodes, * metastasized LNs, and blue circles = tumor-infiltrating immune cells.

**Figure 2 diagnostics-12-01408-f002:**
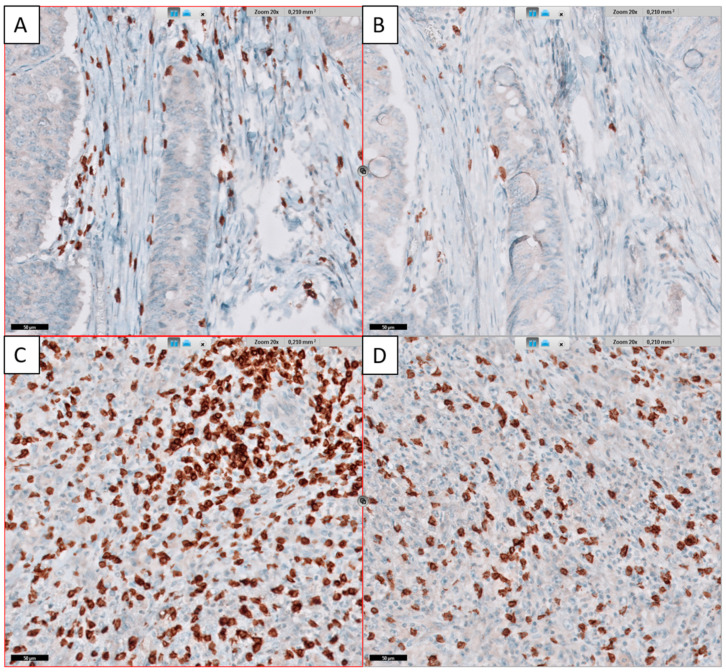
Immunohistochemical staining of tumor-infiltrating lymphocytes in the tumor center. CD3 lymphocytes (**A**) and CD8 lymphocytes (**B**) in a patient with low lymphocyte infiltration. CD3 lymphocytes (**C**) and CD8 lymphocytes (**D**) in a patient with high lymphocyte infiltration. Scale bar indicates 50 µm.

**Figure 3 diagnostics-12-01408-f003:**
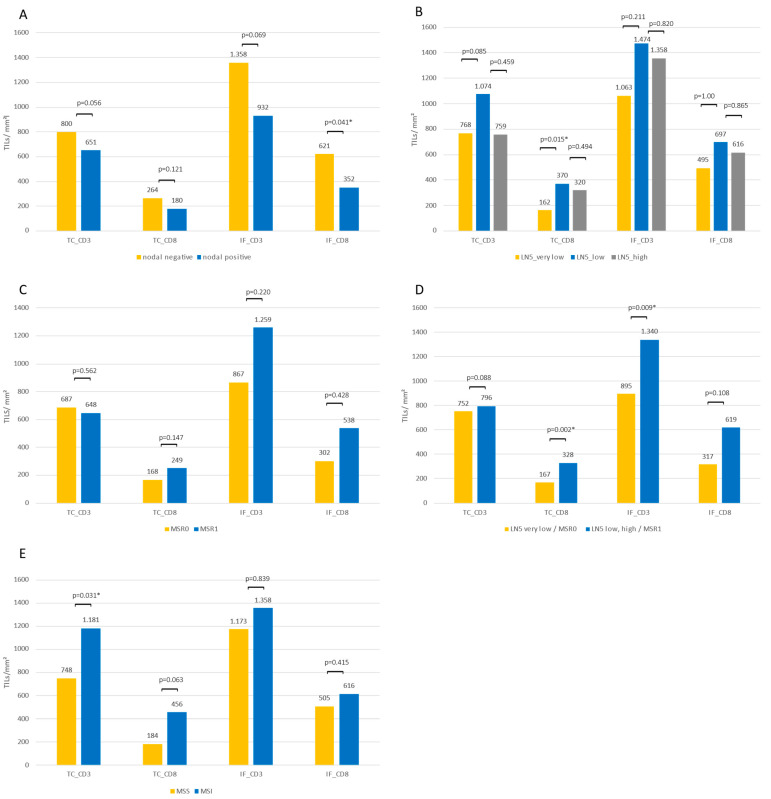
Median values of tumor-infiltrating lymphocytes in (**A**) nodal-negative vs. nodal-positive patients, (**B**) according to the LN5 category, (**C**) according to the MSR category, (**D**) patients with a low (LN5 very low and MSR0) or high (LN5 low and high and MSR1) lymph node immune activation, and (**E**) in microsatellite stable or instable patients. TILs: tumor-infiltrating lymphocytes; TC: tumor center; IF: invasive front. Significant *p*-values are marked with * for *p* < 0.05.

**Figure 4 diagnostics-12-01408-f004:**
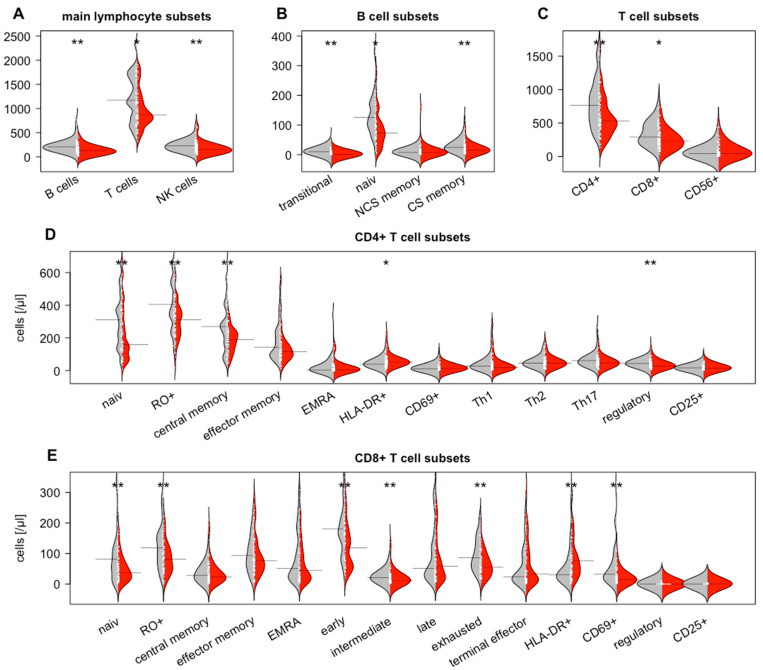
Comparison of the circulating lymphocyte subsets between patients with a low (LN5 very low and MSR0; grey dots) or high (LN5 low and high and MSR1; red dots) lymph node immune activation. (**A**) Main lymphocyte subsets, (**B**) B-cell subsets, (**C**) T-cell subsets, (**D**) CD4+ T-cell subsets, and (**E**) CD8+ T-cell subsets. Significant *p*-values are marked with * for *p* < 0.05 and ** for *p* < 0.005.

**Figure 5 diagnostics-12-01408-f005:**
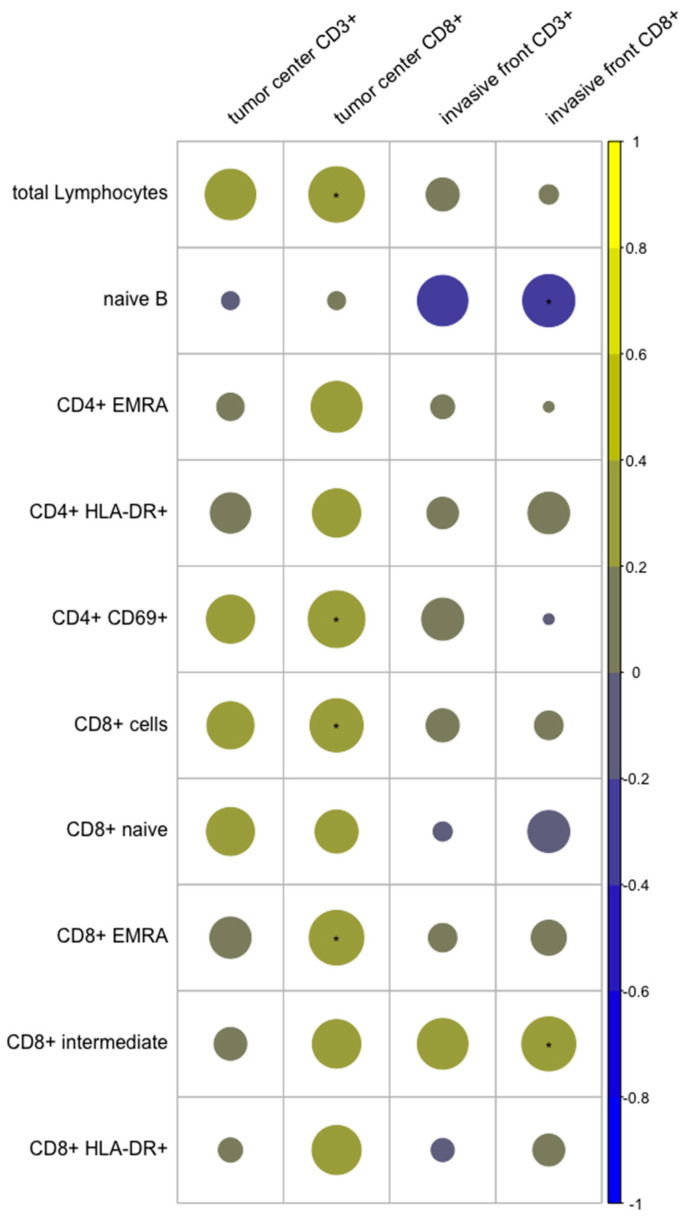
Correlation of tumor-infiltrating and selected circulating lymphocytes. Correlation coefficient on the right side. Positive correlations are displayed in yellow and negative correlations in blue. Asterisks mark significant correlations.

**Table 1 diagnostics-12-01408-t001:** Demographic and disease characteristics.

Variables	Patients (*n* = 47)
Age; median (range)	66 (42–84)
Gender	
male; *n* (%)	29 (62)
female; *n* (%)	18 (38)
Stage	
UICC I; *n* (%)	11 (23)
UICC II; *n* (%)	20 (43)
UICC III; *n* (%)	11 (23)
UICC IV; *n* (*%*)	5 (11)
Number of lymph nodes; median (range)	35 (13–64)
Lymph node infiltration	
yes	16 (34)
no	31 (64)
Tumor sidedness	
right *n* (%)	32 (68)
left *n* (%)	15 (32)
Microsatellite status	
stable *n* (%)	35 (74)
instable *n* (%)	11 (23)
information not available *n* (%)	1 (3)

**Table 2 diagnostics-12-01408-t002:** Peripheral blood and tumor-infiltrating lymphocytes of the entire cohort. Cell counts are given as the median value/µL (interquartile range).

	Colon Carcinoma Patients Median Cell Count (Interquartile Range) (*n* = 47)
Total lymphocytes	1320 (1046–1666)
CD3+ cells	868 (714–1190)
CD8+ cells	229 (131–344)
Naive	36 (16–67)
Memory	82 (45–118)
CM	23 (13–50)
EM	76 (45–112)
EMRA	45 (17–114)
Early	119 (58–169)
Intermediate	11 (6–26)
Late	58 (19–150)
Exhausted	54 (29–89)
Terminal effector	34 (11–117)
HLA-DR+	76 (37–141)
CD69+	15 (8–25)
CD4+ cells	528 (400–768)
Naive	159 (79–295)
Memory	311 (214–378)
CM	190 (124–238)
EM	116 (78–167)
EMRA	5 (2–19)
Th1	19 (10–40)
Th2	46 (32–62)
Th17/Th22	47 (33–67)
CD25high	15 (8–25)
HLA-DR+	52 (44–65)
CD69+	13 (9–21)
NK cells	150 (87–223)
CD56+ CD16+	129 (65–203)
CD56dim CD16bright	9 (7–18)
CD56bright CD16dim	11 (8–14)
NK-like T cells	43 (19–108)
B cells	122 (69–185)
Naive	73 (37–113)
Memory	7 (3–14)
Class switch	16 (8–24)
Transitory	2 (1–4)
CD4/CD8 Ratio	2.2 (1.8–3.3)
Histology	
TILs TC CD3	768 (617–1074)
TILs TC CD8	235 (148–440)
TILs IF CD3	1173 (838–1577)
TILs IFCD8	505 (277–820)

## Data Availability

The data generated in this study are available upon request from the corresponding author.

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
