# Peer review of "Circulating Lymphocytes Reflect the Local Immune Response in Patients with Colorectal Carcinoma"

_diagnostics, 2022, doi:10.3390/diagnostics12061408_

Round 1

Reviewer 1 Report

Waidhauser et al. showed that circulating lymphocytes, specifically cytotoxic T cells, correlate with local anti-tumor immune response exhibited by tumor-infiltrating lymphocytes and lymph node activation. Overall, this research may add value to the literature for the diagnosis/treatment of colorectal cancer (CRC). However, with 47 CRC patients, the sample size is small, and the results may vary with large scale prospective study that the authors mentioned as a limitation of the study. The manuscript may be accepted after the authors address following comments.

In the Introduction, the authors should include few sentences describing the current morbidity and mortality of CRC as well as main diagnostic options in brief.

The authors should add scale bar in Figure 2 showing immunohistochemical staining of tumor-infiltrating lymphocytes in the tumor.

The authors should redraw all the bar diagrams in Figure 3 to improve it. Moreover, the presentation of A, B, C, D….. in Figure 3 and 4 are different. In Figure 3 it is boxed, in Figure 4 no box.

In Table 1, median age is 65,5 years but in the text 66 years. The authors should write the same number in the text and table.

In Table 1, the microsatellite status shows total number of stable and instable patient is 46 (35+11). The authors should include the microsatellite status of remaining 1 patient.

Author Response

Reviewer #1:

Waidhauser et al. showed that circulating lymphocytes, specifically cytotoxic T cells, correlate with local anti-tumor immune response exhibited by tumor-infiltrating lymphocytes and lymph node activation. Overall, this research may add value to the literature for the diagnosis/treatment of colorectal cancer (CRC). However, with 47 CRC patients, the sample size is small, and the results may vary with large scale prospective study that the authors mentioned as a limitation of the study. The manuscript may be accepted after the authors address following comments.

In the Introduction, the authors should include few sentences describing the current morbidity and mortality of CRC as well as main diagnostic options in brief.

A: Thank you for your valuable comments on our manuscript. Regarding morbidity, mortality and diagnostic/ screening we added according sentences to the introduction.

The authors should add scale bar in Figure 2 showing immunohistochemical staining of tumor-infiltrating lymphocytes in the tumor.

A: We build a new Figure 2 including scale bars.

The authors should redraw all the bar diagrams in Figure 3 to improve it. Moreover, the presentation of A, B, C, D….. in Figure 3 and 4 are different. In Figure 3 it is boxed, in Figure 4 no box.

A: We redraw Figure 3 in color and changed the presentation of A,B,C,D to unboxed. We hope you find the presentation better now, if not could you please specify what changes you would prefer. 

In Table 1, median age is 65,5 years but in the text 66 years. The authors should write the same number in the text and table.

A: We changed the presentation in the table to 66 years.

In Table 1, the microsatellite status shows total number of stable and instable patient is 46 (35+11). The authors should include the microsatellite status of remaining 1 patient.

A: For the remaining patient no applicable result regarding MSI status was obtained. This information was added to Table 1.

Reviewer 2 Report

Waidhauser and colleagues describe how the patient response to CRC correlates with circulating lymphocytes in peripheral blood. Overall, the study is well-designed, and the analyses are clearly present. The authors also identify some limitations of the study, such as the need to confirm the tumor-reactiveness of circulating lymphocytes (by analyzing the TCR repertoire, for example). However, these additional experiments could be performed in further studies. I have no specific requirements before publication. I would suggest just to revise the layout of the figure 4, which is quite difficult to read (median bars are too large and violin plot too small).

Author Response

Reviewer #2

Waidhauser and colleagues describe how the patient response to CRC correlates with circulating lymphocytes in peripheral blood. Overall, the study is well-designed, and the analyses are clearly present. The authors also identify some limitations of the study, such as the need to confirm the tumor-reactiveness of circulating lymphocytes (by analyzing the TCR repertoire, for example). However, these additional experiments could be performed in further studies. I have no specific requirements before publication. I would suggest just to revise the layout of the figure 4, which is quite difficult to read (median bars are too large and violin plot too small).

A: Thank you for your valuable comments! We redraw Figure 4 according to your suggestions and hope it becomes easier to read now.